# Non-Additive and Asymmetric Allelic Expression of *p38 mapk* in Hybrid Tilapia (*Oreochromis niloticus* ♀ × *O. aureus* ♂)

**DOI:** 10.3390/ani14020266

**Published:** 2024-01-15

**Authors:** Zihui Liu, Binglin Chen, Zhiying Zou, Dayu Li, Jinglin Zhu, Jie Yu, Wei Xiao, Hong Yang

**Affiliations:** 1Wuxi Fisheries College, Nanjing Agricultural University, Wuxi 214081, China; lzh_quzhou@163.com; 2Key Laboratory of Freshwater Fisheries and Germplasm Resources Utilization, Ministry of Agriculture and Rural Affairs, Freshwater Fisheries Research Center, Chinese Academy of Fishery Sciences, Wuxi 214081, China; chenbinglin@ffrc.cn (B.C.); zouzy@ffrc.cn (Z.Z.); lidy@ffrc.cn (D.L.); zhujl@ffrc.cn (J.Z.); yujie@ffrc.cn (J.Y.)

**Keywords:** hybridization, non-additive, asymmetric allelic expression, cis- and trans-acting elements, *p38 mapk*

## Abstract

**Simple Summary:**

Mitogen-activated protein kinases (MAPKs) are serine/threonine kinases that play a crucial role in regulating multiple processes, including protein degradation and localization, mRNA stability, endocytosis, apoptosis, cytoskeleton dynamics, and cell migration. To investigate the regulation of MAPKs’ gene expression variation in hybrid tilapia (*Oreochromis niloticus* ♀ × *O. aureus* ♂), we focused on *p38 mapk*. Our study uncovered a compensatory interaction between cis- and trans-acting elements of the Nile tilapia and blue tilapia sub-genomes, resulting in a nonadditive expression pattern of *p38* in hybrid tilapia. In addition, we identified eight specific single-nucleotide polymorphisms (SNPs) in transcription factor binding sites between Nile tilapia and blue tilapia and demonstrated that it is promoter differences that differentiate parental gene expression using a dual-luciferase reporting system. Overall, this study reveals the non-additive expression patterns of functionally important genes in hybrid fish, particularly in growth and immune responses.

**Abstract:**

Hybridization is a widely used breeding technique in fish species that enhances desirable traits in cultured species through heterosis. However, the mechanism by which hybrids alter gene expression to form heterosis remains unclear. In this study, a group of hybrid tilapia was used to elucidate heterosis through interspecies crossing. Specifically, *p38* was analyzed to describe the regulation of gene expression variation in hybrid tilapia. Transcripts from the Nile tilapia allele were found to be significantly higher than those from the blue tilapia allele in hybrid individuals, indicating that the expression of *p38* was dominated by Nile tilapia sub-genomic alleles. The study also found a compensatory interaction of cis- and trans-acting elements of the Nile tilapia and blue tilapia sub-genomes, inducing a non-additive expression of *p38* in hybrids. Eight specific SNPs were identified in the *p38* promoter regions of Nile tilapia and blue tilapia, and were found to be promoter differences leading to differences in gene expression efficiencies between parental alleles using a dual-luciferase reporter system. This study provides insights into the non-additive expression patterns of key functional genes in fish hybrids related to growth and immunity response.

## 1. Introduction

Interspecific hybridization is very common in plants and animals. The fusion of two genomes may lead to new phenotypic variants, creating heterosis in hybrids that ensures their survival advantages compared with parents [1]. Heterosis results in superiority in growth potential, stress tolerance, yield, and quality that exceeds the two parents [2]. Heterosis can be characterized by the ability to produce faster growing offspring. For example, the triploid crucian carp (3*n* = 150), formed by crossing allotetraploid crucian carp (4*n* = 200) as the male and Japanese white crucian carp as the female, is characterized by fast growth rate, sterility, and excellent meat quality [3,4]. Heterosis can be characterized by the ability to produce more adaptable offspring. For example, Qinglong grouper (*Epinephelus coioides* ♀ × *E. lanceolatus* ♂) has a high survival rate and disease resistance [5]. In tilapias, compared to Nile tilapia (*Oreochromis niloticus*) and blue tilapia (*Oreochromis aureus*), the hybrid tilapia (*Oreochromis niloticus* × *Oreochromis aureus*) has several advantages, such as high male ratio, rapid growth, and good disease resistance, which make it a leading commercially cultured tilapia species in China [6,7]. However, how the hybrid tilapia achieves these hybrid vigors is still unknown. Therefore, understanding the molecular mechanisms of heterosis is important for species formation and hybrid breeding.

Hybridization affects the individual gene expression that results in the success of hybrid varieties [8,9]. Compared to the parents, the gene expression of the hybrid progeny can be divided into additive and non-additive gene expressions [10]. Non-additive gene expression is one of the expression patterns of heterosis [11]. In hybrids, the expression of non-additive genes can be determined when their expression is not equal to the mid-parental value (MPV). In addition to the overall expression pattern, allele-specific expression (ASE) has been suggested as another mechanism of hybrid vigor [12]. Alleles from both parents on the same locus sometimes show significant differences in expression, a phenomenon known as allele-specific expression. Allele-specific expression is a very interesting feature, and it is considered to be one of the important sources of biological phenotypic variation [13]. In agricultural breeding, allele-specific expression is often used to explain the mechanisms of dominant trait formation in plant hybrids such as rice [12], maize [14], and cabbage [14]. It has been shown that gene expression changes in ASE are mediated by cis- and/or trans-regulatory changes [15]. Cis- and/or trans-regulatory changes can be identified in interspecific hybrids by analyzing their expression characteristics [16].

In the study of fish, it has also been found that allele-specific expression is involved in the sex differentiation and body color formation of individuals in the hybrid offspring. In individuals of the Amazon molly (*P. mexicana* × *P. latipinna*), the expression of the allele *Arα* was significantly higher in Atlantic mollies than in sailfin mollies, indicating that the biased expression of *Ara* in hybrid individuals might be closely related to the sexual differentiation of Amazon mollies [17]. It was found that hybrid individuals of red swordtail (*Xiphophorus helleri*) and spotted swordtail (*X. maculatus*) specifically expressed the red swordtail allele in a 5.81 Mb long segment on chromosome group LG5, and this ASE phenomenon might be closely associated with somatic melanoma formation in hybrid individuals [18]. In hybrid tilapia (*Oreochromis niloticus* × *O. aureus*), it was found that the mRNA expression of the growth hormone (GH) in the pituitary gland was significantly higher than that of both parents and showed asymmetric expression at the level of allelic transcripts. The ASE values of the hybrid tilapia specifically expressed the Nile tilapia GH allele and promoted the expression advantage of GH in the hybrid tilapia [19]. The above studies indicate that allele-specific expression in hybrid offspring contributes to key gene expression regulation, which has important research implications for us as we improve the theory of heterosis and use these theories to guide the breeding of new fish species with greater superiority.

Mitogen-activated protein kinases (MAPKs) are a class of serine/threonine kinases mainly composed of three families: extracellular signal-regulated kinase (ERK), c-Jun N-terminal protein kinase (JNK), and *p38* MAPK [20]. ERKs mainly act as a central component of the signal transduction pathway and are regulated by relevant growth stimulators. JNK is activated in response to stimuli such as high osmotic pressure or strong oxidative conditions, and *p38* is involved in a variety of physiological processes such as inflammation, cell growth, cell differentiation, cell cycle, and cell death [21]. In fish such as rock bream (*Oplegnathus fasciatus*) [22], Atlantic salmon (*Salmo salar*) [23], and rainbow trout (*Oncorhynchus mykiss*) [24], previous studies have focused on comparing the sequence structure and examining the differences in the stress expression of MAPK genes. These studies have made some progress and helped us to understand the mechanism of MAPK’s role in fish. In our previous study, transcriptome analysis revealed that *p38* showed non-additive expression in hybrid tilapia and played a key role in multiple enriched pathways such as the endocrine system, disease immunity, and the oxidative metabolism [25,26]. Hu et al. [3] found that, in Nile tilapia and hybrid tilapia, *S. agalactiae* infection can activate the *p38* MAPK pathway, and the rapid response and high level of protein expression of *p38* MAPK may be the reason why hybrid tilapia have stronger bacterial resistance than Nile tilapia. Therefore, in this study, we analyzed the sequence and assessed the allele-specific expression pattern of *p38* in hybrid tilapia, and explored the regulation of *p38* expression by cis- and trans-acting elements in hybrid tilapia with a view to demonstrating that *p38* can promote fish growth.

## 2. Materials and Methods

### 2.1. Ethics Statement

All experimental protocols were approved by the Animal Care and Use Committee of Nanjing Agricultural University (SYXK-2021-0086, Nanjing, China). Fish were maintained in well-aerated water and were anesthetized with MS-222 before sampling, and their livers were extracted following the Guidelines for the Care and Use of Laboratory Animals in China.

### 2.2. Rearing of Experimental Fish

Blue tilapia (AR) and Nile tilapia (NL) came from the Freshwater Fisheries Research Center, Chinese Academy of Fishery Sciences. In June 2021, adult blue tilapia ♀ × ♂, Nile tilapia ♀ × ♂, and Nile tilapia ♀ × blue tilapia ♂ were bred in pairs. Adult male and female fish were selected from each species and placed in glass tanks for two weeks, with water temperatures maintained at 27–28 °C, ammonia nitrogen maintained below 0.4 mg/L, and dissolved oxygen maintained above 6.0 mg/L. Commercial diets with 0.03% vitamin E were fed during this period. Adult fish were injected intraperitoneally with mixed oxytocin (female: 5.0 mg/kg domperidone, 2500 IU/kg gonadotropin, 10.0 μg/kg luteinizing hormone-releasing hormone A2; male: half the dose of females; Ningbo Second Hormone Factory, Ningbo, China). Eight hours after injection, we collected eggs and sperm from females and males by gently squeezing the abdomen. Stocks of AR ♀ × ♂, NL ♀ × ♂, and NL ♀ × AR ♂ (HY) were artificially fertilized. We fertilized matured eggs with sperm in culture dishes and stirred them for 2 min. Then, fertilized eggs (AR, NL, HY) were poured into an incubator and maintained for seven days at 27 °C to obtain fry that swam typically. Three hundred successfully hatched fries from each group were placed in circular tanks and fed with a powdery meal (20 μm pore size) in a bait station three times a day (6:00, 12:00 and 18:00); the powdery meal contained 50.0% crude protein, 4.0% crude fat, 4.0% crude fiber, 18.0% crude ash, 2.0–5.0% calcium, and 1.0% phosphorus. After two months of culture, 50 male individuals were randomly selected from each group, then placed in 60 m^3^ ponds for culture. We maintained a water temperature range of 27–32 °C and an average daily photoperiod at 13 h of light and darkness followed by 11 h of rest. In addition to the temperature and photoperiod, we closely monitored the concentration of total ammonia nitrogen (TAN) and nitrite in the water. TAN concentration was kept below 0.5 mg/L, while nitrite concentration was maintained below 0.3 mg/L. We ensured that the pH value remained in a range of 7.0–8.0 and dissolved oxygen levels remained above 6.0 mg/L. During the culturing process, 2 mm diameter floating feeds (Ocialis, ADM Animal Nutrition Co., Ltd., Nanjing, China) were fed daily with 30.0% crude protein, 4.0% crude fat, 10.0% crude fiber, 13.0% crude ash, 0.3–1.8% calcium, 1.0% phosphorus, and 0.1–1.5% sodium chloride. Artificial monitoring was conducted for thirty minutes to ensure that individuals ate until they were satisfied. Any leftover food was promptly removed.

### 2.3. Sample Collection and Analysis

At 45 d of rearing, we restricted feeding for 24 h. We randomly collected twenty individuals from each group (NL: 133.0 ± 16.8 g, AR: 158.5 ± 29.0 g, HY: 226.1 ± 27.2 g). After light anesthesia using 20 mg/L MS-222, we dissected individuals and collected liver tissue with liquid nitrogen for gene expression studies. Moreover, twenty individual caudal fins of each group were clipped to an area of approximately 1 cm^2^ and used to extract genomic DNA.

### 2.4. Extraction and Quality Control of Total RNA and DNA

Total RNA was extracted from tilapia liver tissue using TRIZOL reagent (Invitrogen, Waltham, MA, USA). Genomic DNA was removed using DNase-I (TaKaRa, Kyoto, Japan). Then, 1.0% agarose electrophoresis was used, the RNA concentration of each sample was determined using NanoDrop-2000 (Thermo Fisher, Waltham, MA, USA), and high-quality RNA samples were selected (OD260/280 = 1.8–2.2, OD260/230 > 2.0, RIN > 7) to construct the cDNA library. The mRNA was reverse-transcribed into cDNA using a reverse transcription kit (Vazyme, Nanjing, China): HiScript^®^ RT SuperMix4 μL, 1.0 μg RNA, RNase-free water to 20 μL; the reaction procedure was maintained at 37 °C for 15 min followed by denaturation at 85 °C for 5 s. Genomic DNA was extracted from the tail fin of tilapia using universal animal genomic DNA extraction kits (TIANGen, Beijing, China).

### 2.5. Confirmation of SNP Loci in the Coding Region of p38

We searched for SNP loci on the cDNA of 20 individuals between NL and AR. The primers were designed in the coding region of the *p38* gene based on the tilapia genome (https://www.ncbi.nlm.nih.gov/assembly/GCA_001858045.3, accessed on 21 April 2021) using Primer Premier 5.0 (Premier, Markham, ON, Canada; Table 1).

We performed the amplification reactions for *p38* using a 50 μL volume: 25.0 μL of 2× Plus Dye Premix Taq (TaKaRa, Japan), 0.4 μL of each forward and reverse primer (10 μM), 1.0 μL of cDNA template, and 8.2 μL of RNase-free water. We set the PCR reaction procedure as follows: initial denaturation at 94 °C for 5 min; 95 °C for 30 s; 55–60 °C for 30 s; 72 °C for 60 s; 35 cycles; 72 °C for 10 min. The amplified products were purified with 1.5% agarose gel electrophoresis and sent to Shanghai Sangon Biotech for Sanger sequencing. The sequences were compared by Snapgene 6.0 (Dotmatics, San Diego, CA, USA) screening for specific SNP loci between the NL and AR and reconfirmation in their genomic DNA. Specific SNP loci between two NL and AR were confirmed in the cDNA and genomic DNA of HY with the same primers.

### 2.6. Specific Expression of p38 Allele in the Hybrid Tilapia

According to the methodology by Coolon et al. [16], an allele-specific expression bias study was carried out on the cDNA of the hybrid tilapia (HY). Using the SNP locus as a site-specific locus, cDNA samples from 20 HY were collected and commissioned to Shanghai Sangon Biotech for pyrophosphate sequencing to observe the specific expression differences of different alleles from the subgenome of Nile and blue tilapia. The analysis process mainly consisted of amplifying SNP differential site fragments from HY liver cDNA samples (Table 1) and purifying the products using 1.5% agarose gel electrophoresis. Then, the purified products were sequenced by pyrosequencing on a PyroMark Q96 ID instrument (Qiagen, Germantown, MD, USA). The subgenomic mRNA expression ratios of *p38* allele expression bias in HY individuals were analyzed.

### 2.7. Quantitative Analysis of p38 Expression among Different Strains

Real-time quantitative PCR (qRT-PCR) was performed on *p38* in the liver tissue of NL, AR, and HY. Each group had three replicates and β-actin was used as a quantitative reference gene. The primers were designed by Primer Premier 5.0 (Premier, Markham, ON, Canada; Table 1). Real-time PCR was performed on an ABI 7900HT instrument (Applied Biosystems, Foster City, CA, USA). The qRT-PCR reaction volume was 20 μL: 10 μL of ChamQ SYBR qRT-PCR Master Mix (Vazyme, Nanjing, China), 0.4 μL of each forward and reverse primer (10 μM), 1.0 μL of cDNA template, and 8.2 μL of RNase-free water. The reaction procedure was set as follows: initial denaturation at 95 °C for 30 s; 95 °C for 10 s; 60 °C for 30 s; 40 cycles. This was followed by a melting curve analysis (95 °C for 10 s, 60 °C for 60 s, and 95 °C for 15 s). The transcript levels of the target gene were analyzed by the 2^−ΔΔCt^ method. All the data are expressed as mean ± SD, with *p* < 0.05 representing significant differences.

### 2.8. Cis-Trans Effects from the Parents in Hybrid Tilapia

According to Shi et al.’s [27] model of allele expression analysis, the allelic cis–trans effects of *p38* alleles from Nile tilapia and blue tilapia were calculated in hybrid tilapia individuals, as shown in Figure 1.

### 2.9. Analysis of Differences in the Gene Promoter Region

The promoter region sequences of *p38* of NL, AR, and HY were amplified according to tilapia genome sequence information (https://www.ncbi.nlm.nih.gov/assembly/GCA_001858045.3, accessed on 21 April 2021). The amplification interval was set between −2000 and −1 upstream of the transcription start site (TSS) [22]. To analyze the polymorphism of the promoter region of *p38*, we designed three pairs of specific primers, as listed in Table 2. We performed the amplification reactions for the promoter region of *p38* using a 50 μL mixture: 25 µL of Plus DyePremix Taq (TaKaRa, Japan), 1 µL of each forward and reverse primer (10 µM), 2 µL of DNA template, and RNaes-free water added up to 50 µL. We set the PCR reaction procedures as follows: initial denaturation at 94 °C for 5 min; 95 °C for 30 s; 55–60 °C for 30 s; 72 °C for 60 s; 35 cycles in total; 72 °C for 10 min. The amplified products were purified with 1.5% agarose gel electrophoresis and sent to Shanghai Sangon Biotech for Sanger sequencing. The sequences were compared by Snapgene 6.0 (Dotmatics, San Diego, CA, USA) screening for specific SNP loci. We used Promoter-2.0-Prediction-Server (http://www.cbs.dtu.dk/services/Promoter/, accessed on 22 September 2021) to predict the core promoter region of the 5′ flanking region of the *p38* gene. The functions of SNP loci in the promoter region were identified using MatInspector and the motif program in Primer Premier 5.0 software. SNPs that could alter transcription factor binding sites (TFBS) were retained.

### 2.10. Dual-Luciferase Activity Assay of SNP Site Promoters

According to the methodology by Zhou et al. [28], pGL3 recombinant vectors for AR and NL were constructed using the principle of homologous recombination, and SNP promoters of growth-related genes in the p38 pathway were detected. Homologous recombinant primers were designed and amplified by PCR. The forward primer sequence was CGATCTAAGTAAGCTAGAACAGTCCAGCTCCTCTCTTTG and the reverse primer sequence was CCGGAATGCCAAGCTTTGGGCTCAAGAACAATCATGAGTG. The integrity of PCR products was tested using 1.0% agarose gel electrophoresis. The pGL3-Basic vector was linearized by HindIII fast Restriction endonuclide (Takara, Japan). The two fragments were homologous recombined with homologous recombinase (Takara, Japan). After the recombination was completed, the plasmid was sequenced to confirm the correctness.

HEK293T cells were cultured in DMEM medium containing fetal bovine serum (Dulbecco’s Modified Eagle Medium, DMEM) (TransGen, Beijing, China), and transfected every 24 h when the cell confluent degree reached 70%. Transfection was performed with TransGen EL Transfection Reagent (TransGen, Beijing), and the recombinant plasmid identified correctly by sequencing was transfected into 293T cells cultured in 24-well plates. Twenty-four hours after transfection, following the instructions of the dual-luciferase assay kit (TransGen, Beijing), the fluorescence ratio between firefly and sea kidney in each group was calculated, and promoter activity was calculated.

## 3. Results

### 3.1. Gene p38 Coding Region and SNP Locus Screening

The analysis revealed that the gene *p38* coding region is 1086 bp long and distributed over 12 exons (Figure 2A), encoding a total of 361 amino acids. A total of seven SNP loci were identified in the three groups by segmental amplification and multiple comparisons of the *p38* coding regions of AR, NL, and HY (Figure 2B). Using the transcription start site as +1, the positions and base information of these seven SNP sites were S1 (g. 325 G > A), S2 (g. 601 C > T), S3 (g. 817 C > T), S4 (g. 841 A > G), S5 (g. 1142 C > T), S6 (g. 1270 C > T), and S7 (g. 1357 G > A), respectively. Among them, SNPs were present at S2, S5, and S6 within AR. SNPs were present at S1, S2, S3, S5, and S6 within NL, and SNPs were present at all loci in HY. Furthermore, notable variations existed between NL and AR concerning S4 and S7. In S4 (Figure 3A), NL exhibited genotype A, while AR had genotype G, and HY showed a combination of genotypes A/G; in S7 (Figure 3B), NL exhibited genotype G, while AR exhibited genotype A, and HY showed a combination of genotypes G/A. In addition, all seven loci were synonymous mutations, and all SNP mutations did not alter the amino acid sequence encoded by the gene *p38*.

### 3.2. Gene p38 Allele-Specific Expression

Pyrophosphate sequencing was carried out using S7 (g. 1357 G > A) as the locus, and the *p38* allele expression frequency in HY showed that alleles from NL and AR showed a bias in expression in liver tissue (Figure 4; Appendix A). The number of mRNAs of the alleles from NL (G) to the expression frequency ratio of the alleles from AR (A) in HY individuals was 77:23, and the expression level of the alleles from NL was much higher than that of the alleles from AR. HY individuals were mainly dominated by the alleles from NL in *p38* and the cis effect obtained for the NL/AR allele ratio within HY was 1.74.

### 3.3. Cis and Trans Effects between p38 Alleles in HY

The *p38* expression levels of NL, AR, and HY liver tissues are shown in Figure 5. All three groups were significantly different from each other (*p* < 0.05), and the *p38* expression levels were HY > NL > AR in descending order, where the mean values of HY and NL expression were 1.90-fold and 1.48-fold those of AR, respectively, and the *p38* expression levels of HY were significantly higher than those of both parents (*p* < 0.05), showing an ELOD-Up (Expression Level Over Dominance—Up) expression pattern. Furthermore, the combined cis and trans effect was determined to be 0.92 by assessing the relative expression levels of *p38* between NL and AR, and combined with the cis effect value obtained from the NL/AR expression ratio of *p38* by pyrophosphate sequencing in HY, the trans effect value of *p38* was −0.98. A compensatory interaction between cis and trans factors was observed, leading to a non-additive expression of *p38* in HY individuals.

### 3.4. Distribution of SNP Sites in the 5′ Flanking Region of p38 Gene and Prediction of Transcription Factor Binding Sites

A promoter region SNP site analysis was performed on the upstream sequence of the 5′ flanking region using the transcriptional start site marker of *p38* as +1. All NL and AR individuals had a sequence identity in the 5′ flanking region of *p38*, but there were eight SNP loci between the groups (Figure 6A), namely S’1 (g. −314 TTT > ---), S’2 (g. −297 C > T), S’3 (g. −241 G > C), S’4 (g. −238 C-- > AAG), S’5 (g. −236 G > A), S’6 (g. −145 T > -), S’7 (g. −138 T > A), and S’8 (g. −33 T > A), and the location distribution and base information of these SNP sites in the promoter region are shown in Figure 6B. Analysis by prediction software identified a core promoter element in the 5′ flanking region of *p38*, specifically at −100 bp. Two SNP loci, known as S’3 and S’7, were also examined for potential variations in transcription factor binding sites (Figure 6A). Notably, at the S’3 locus, NL exhibited an additional transcription factor Hoxb6 binding site (TTAAGTAC), compared to AR. Similarly, at the SNP site within the S’7 locus, NL had an extra transcription factor Dlx2 binding site (TTAATTTT) compared to AR. The allele-specific expression observed in HY from different parental sources may be due to SNPs in the promoter region associated with changes in allele expression in hybridization.

### 3.5. Promoter Relative Activity Detection

Dual-luciferase activity assays were performed on NL and AR, and the results showed (Figure 7) that the promoter activity of *mapk14a* in NL was significantly higher than that of AR (*p* < 0.05), and the promoter activity of NL was 3.3 times higher than the promoter activity of AR.

## 4. Discussion

### 4.1. The Role of Coding Region SNP Markers in Tilapia p38 Allele-Specific Expression Studies

SNPs are a common genetic molecular marker developed in recent years, and have the characteristics of strong heritability and high stability. Initially, the study of SNPs was aimed at understanding their potential for sequence alteration. However, as molecular markers and their detection techniques advance, the perspective has shifted to considering the potential impact of SNPs on gene regulation, such as through methylation processes. Therefore, SNPs are also widely used in various genetic research fields. In fish, SNPs have emerged as valuable tools for molecular marker-assisted selection, due to their ability to identify specific SNP loci associated with traits like growth and resistance.

Li et al. [29] screened five SNP loci in the 5′ flanking region of the muscle growth inhibitor gene *mstn* in spotted halibut (*Verasper variegatus*), and found that individuals with the CC genotype at the g. −355 T > C locus had higher growth trait phenotypes, indicating that polymorphism at this locus is involved in participating in the regulation of *mstn* gene expression. Thi et al. [29] detected a total of seven SNP loci on the insulin-like growth factor *Igf-1* and receptor *Igf-1R* genes of striped catfish (*Pangasianodon hypophthalmus*), and found that the AA genotype at the g. 13680 A > T locus of the *igf-1* gene and the TT genotype at the g. 13357 T > C locus of the *Igf-1R* gene had a positive cumulative effect on growth rate and could promote the growth rate of the threadfin eel catfish. In tilapia, Chen et al. [30] detected seven SNP loci in the Nile tilapia appetitive peptide prepro-orexin gene, including the GG genotype at the g. −1108 C > G locus, the TT genotype at the g. −1063 C > T locus, and the TT genotype at the g. −883 C > T locus. Genotypes had a positive cumulative effect on growth rate and were presumably closely related to the growth and feeding behaviors of Nile tilapia. In the present study, we identified two specific SNP loci in the coding region of *p38* that could distinguish Nile tilapia and blue tilapia. There are two sides to consider in this matter. Each of these examples identifies specific SNPs associated with growth and resistance by means of SNP molecular markers, which can be useful in fish breeding. Firstly, this method serves as a valuable technical tool for distinguishing between blue tilapia and Nile tilapia fry, which is crucial in preventing the mixing of different species. On the other hand, it also offers a molecular marker to aid in the study of the specific expression of *p38* alleles in hybrid tilapia. Earlier research has made significant advancements in the examination of allelic expression bias in fish, including hybrid yellow catfish and hybrid carp. These studies provided valuable data that can be used to investigate phenotypic traits and analyze the heterosis of these fish hybrids [31,32]. Consequently, this study aimed to investigate the specific expression of hybrid tilapia alleles by employing inter-*p38*SNP markers. This method is helpful for us to reveal the heterosis of hybrid fish in terms of growth and disease resistance.

Some studies have shown that certain synonymous single-nucleotide polymorphism (SNP) mutations can impact the function of codons. In organisms’ genomes, G/C-terminated synonymous codons tend to be more prevalent than expected by chance, indicating a preference for G/C-terminated codons [33]. Consequently, the tRNAs that correspond to G/C-terminated synonymous codons are decoded at a higher rate during mRNA translation extension [34]. Although the SNPs at two loci, S4 and S7, between Nile tilapia and blue tilapia species are synonymous mutations that do not alter the mRNA-encoding amino acids of *p38*, these mutations may still influence the regulation of downstream gene expression related to *p38*. Hwang et al. [35] found a synonymous SNP locus (g. 1575 A > G) in the pgm1 gene of four domestic pig breeds (Berkshire, Duroc, Landrace and Yorkshire), and although all encoded leucine, individuals of two of the genotypes showed significant differences in backfat thickness, drip loss, protein content, fat content and shear force, which are important factors affecting meat quality. Wang et al. [36] identified that synonymous SNPs can regulate protein expression through the alteration of microRNA binding sites based on a large-scale analysis of human genomic data. Consequently, it is possible that synonymous SNP mutations in *p38* may also impact the efficiency of downstream protein synthesis.

### 4.2. Role of Cis-Acting Element Variants in p38 Allele-Specific Expression

Transcription acts as a bridge between genotype and trait [37], and is influenced by various factors such as environmental conditions, epigenetic regulation, and transcription factors [38]. Recent advancements in sequencing technology have allowed researchers to delve deeper into the study of allelic differential expression in biological hybrids. High-throughput techniques have revealed a significant number of allele-specific expressions (ASE) in plants and animals, confirming the considerable impact of cis-acting element variations on allelic expression differences [39,40,41]. Consequently, it is hypothesized that single-nucleotide polymorphism (SNP) variants at specific loci (S’3 and S’7 sites) may induce changes in cis-acting elements within the promoter region of *p38* in Nile tilapia and blue tilapia.

In a previous study conducted by Xu et al. [42], it was found that hybrid tilapia inherited all of the lactate dehydrogenase isoenzyme bands from parent Nile tilapia and only some of the lactate dehydrogenase isoenzyme bands from parent blue tilapia during electrophoresis. Similarly, Zhong et al. [19] observed that GH genes exhibited asymmetric expression in hybrid tilapia, with the expression of the maternal sub-genome from Nile tilapia dominating in hybrid tilapia under both feeding and fasting conditions. Our results are consistent with these findings, as *p38* in hybrid tilapia was primarily influenced by the allele from the Nile tilapia. This suggests that certain differentially expressed genes in hybrid tilapia exhibit biased expression towards the maternal Nile tilapia sub-genome. Further exploration of these genes that display biased expression could provide insights into why hybrid tilapia tend to resemble Nile tilapia in terms of growth, metabolism, and other traits.

### 4.3. Effect of Cis and Trans Compensation Effect on the Non-Additive Expression of p38 Gene

Quantitative changes in gene expression are directly regulated by cis-acting and trans-acting elements. In Drosophila hybrid individuals, Wittkopp et al. [43] discovered that variants in cis elements affect allele-specific expression, while variants in trans element impact the expression of both alleles in diploid cells. When the differences in allelic regulation are solely due to changes in cis-acting elements, the allelic expression is expected to be additive. However, numerous genes exhibit non-additive expression patterns in hybrids [44]. This is because the hybridization process exposes the parental allele to trans-acting factors from both parents, resulting in non-additive expression patterns regulated by trans-acting factors. Studying the interaction mechanisms between cis- and trans-acting elements in hybrid individuals from different parents provides new insights. In our study, we observed that Nile tilapia dominate *p38* allele-specific gene expression in hybrid tilapia. Furthermore, the combined effects of cis and trans elements (cis and trans) for *p38* in individuals of Nile tilapia and blue tilapia were only 0.92, with a negative value for the trans effect (A–B). This indicates a compensatory effect of *p38* alleles from Nile tilapia females and blue tilapia males in hybrid tilapia individuals. Among non-conserved expressed genes, compensatory cis and trans interactions were more prevalent than enhanced cis and trans interactions in hybrids. This may be because enhanced cis and trans interactions increase differential gene expression and promote disruption. On the contrary, compensatory cis and trans interactions reduce gene expression differences, as the cis effect is compensated by the opposing effect of the trans element, enhancing the stability of gene expression across species, which is more common in natural selection in biological evolution. Combes et al. [45] conducted a transcriptome analysis of cross offspring from two coffee varieties, *Coffea canephora* × *C. eugenioides*, and discovered that 52% of the annotated genes exhibited allele-specific expression (17% were regulated by both cis- and trans-acting elements). Among the differentially expressed genes between the parents and offspring, 77% showed non-additive expression, including ELD and ELOD patterns. The trans-acting factor of *Coffea canephora* significantly up-regulated the alleles of *C. eugenioides*, revealing the impact of cis and trans elements from different parents on the expression pattern of hybrid genes. The mechanism can also be applied in the study of non-additive expression in hybrid tilapia, as the compensatory effects of cis- and trans-acting elements from males and females can determine the genetic pattern of *p38* ELOD expression in hybrid tilapia. Overall, our understanding of the differential expression mechanism of *p38* in hybrid tilapia is still incomplete. In the future, multiple technical tools will be required to thoroughly explore the regulatory relationship between the cis- and trans-acting elements of *p38*.

## 5. Conclusions

In this study, we examined the gene sequences of *p38* in Nile tilapia, blue tilapia, and their hybrids. We identified seven SNP loci in the coding region of the gene. Two specific SNP loci, S4 (g. 841 A > G) and S7 (g. 1357 G > A), were identified in Nile tilapia and blue tilapia. Using pyrophosphate sequencing, we observed a bias in allele expression, with Nile tilapia alleles being predominantly expressed. Furthermore, we found a compensatory interaction between the cis- and trans-acting elements of the Nile tilapia and blue tilapia sub-genomes in *p38* transcription. This interaction led to non-additive expression patterns of *p38* in hybrid individuals. To explore the molecular mechanisms behind these non-additive expression patterns, we conducted a comparative analysis of the 5′ flanking region sequences of *p38*. Our analyses revealed eight specific SNP loci between Nile tilapia and blue tilapia and uncovered the reasons why promoter differences lead to different gene expression efficiencies between parental alleles using a dual-luciferase system. These findings have significant advantages for investigating hybrid tilapia heterosis, particularly in terms of enhanced growth and immune capabilities.

## Figures and Tables

**Figure 1 animals-14-00266-f001:**
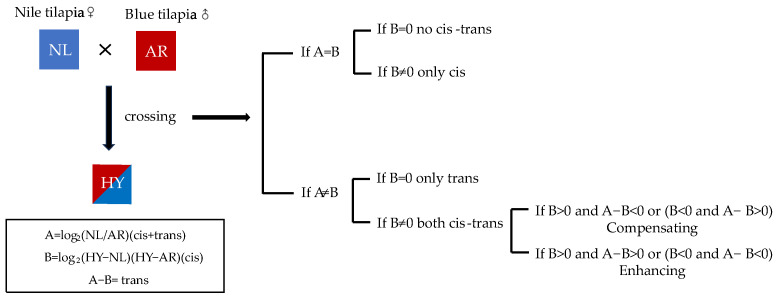
Classification of the transcripts with cis and trans effects in hybrids.

**Figure 2 animals-14-00266-f002:**
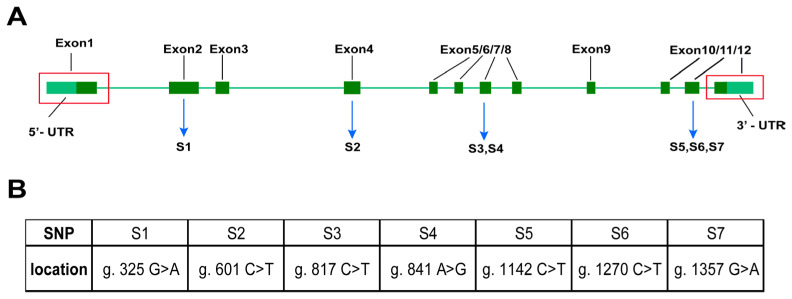
Sequence structure and SNP loci in the coding region of *p38* in tilapia. (**A**) SNP loci in the coding region of *p38* in tilapia. (**B**) The information of specific SNP loci.

**Figure 3 animals-14-00266-f003:**
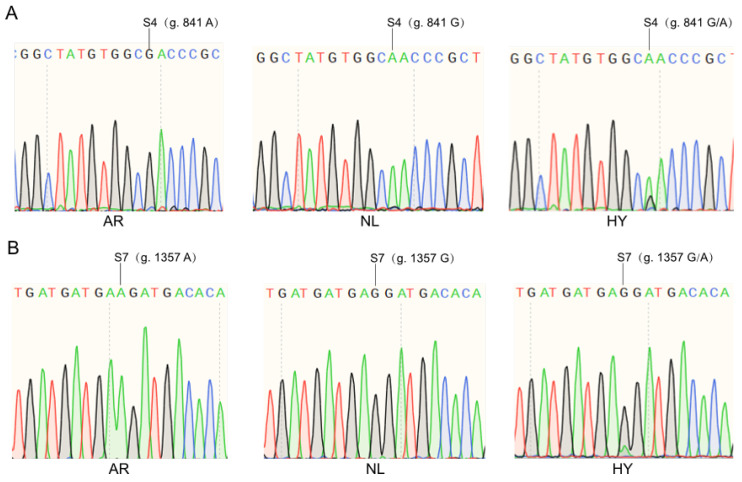
Identification of SNP loci in coding regions of *p38* among NL, AR, and HY. (**A**) Genotypes of S4 on NL, AR, and HY; (**B**) genotypes of S7 on NL, AR, and HY.

**Figure 4 animals-14-00266-f004:**
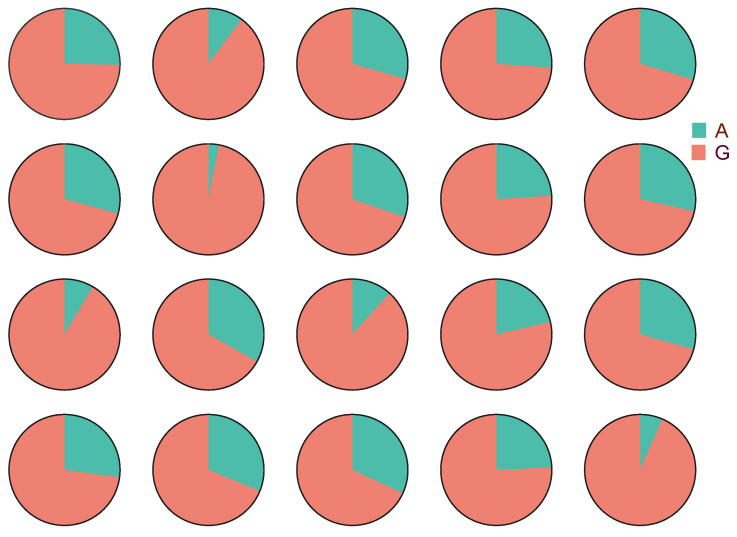
ASE analysis of G (NL allele)/A (AR allele) ratios in *p38* of HY individuals.

**Figure 5 animals-14-00266-f005:**
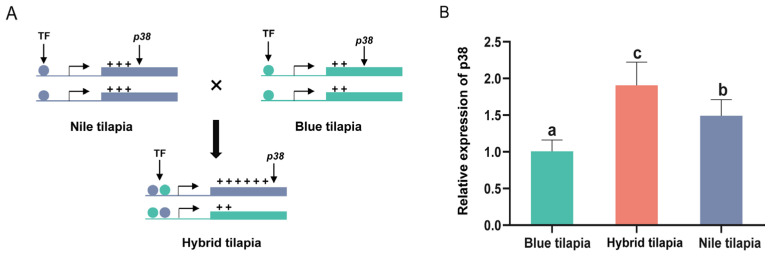
Relative expression of *p38* in the livers of NL, AR, and HY. Different letters indicate significant difference among the groups (*p* < 0.05). ‘+’ indicates the expression of the gene, and more ‘+’ symbols mean higher expression in (**A**). TF: transcription factor. (**B**) is the proportionality between specific expressed quantities.

**Figure 6 animals-14-00266-f006:**
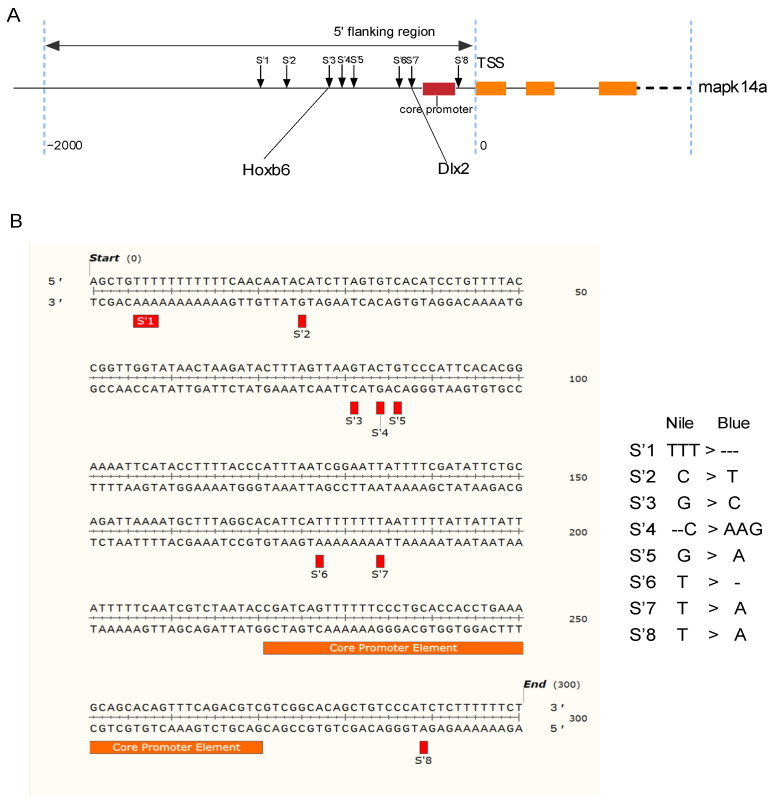
Sequence structure and SNP loci in the 5′ flanking region of *p38* between NL and AR. (**A**) Sequence structure of the 5′ flanking region of *p38* between NL and AR. (**B**) SNP loci in the 5′ flanking region of *p38* between NL and AR with site-specific information to the right.

**Figure 7 animals-14-00266-f007:**
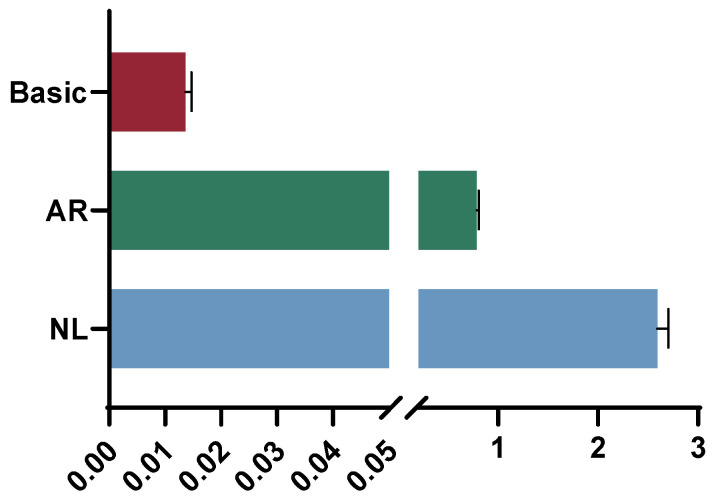
Dual-luciferase activity of promoter structures of *p38*. NL: Activity of *mapk14a* promoter structure in Nile tilapia; AR: activity of *mapk14a* promoter structure in blue tilapia.

**Table 1 animals-14-00266-t001:** The primers for coding regions of *p38*.

Gene	Primers Sequence (5′-3′)	Annealing Temperature	Use	Product Length
*p38-P1-F*	GCGCCGTTTAGCGAAGCGA	59 °C	cDNA	731
*p38-P1-R*	TGTATGATGCCTGCTGAGTGA			
*p38-P2-F*	GACTCATCTAATGGGGGCAG	57 °C	cDNA	752
*p38-P2-R*	CATCATCAAAGCTAGGTGGCT			
*p38-P3-F*	CTGAAGCCGAGCCTTATGAC	60 °C	cDNA	888
*p38-P3-R*	CTTAAGGGATCTTTCGATTGC			
*p38-F*	ACATCAGCTCCTTGCCACACA	60 °C	cDNA	135
*p38-R*	GTGAGCTTCAGCTGCCGTTATC			
*β-actin* *-F*	CTCTGGTCGTACCACTGGTATCG	62 °C	cDNA	169
*β-actin* *-R*	GAAGGAGTAGCCACGCTCCTC			
*p38-PS-F*	GCTCAATACCACGATCCAGAC	55 °C	Pyrophosphate sequencing	193
*p38-PS-BR*	AAAAGCTTGGGTTGAACTGGT			
*p38-P2-FS*	AGCCACCTAGCTTTGATGATG			

**Table 2 animals-14-00266-t002:** The primers for 5′ flanking region of *p38*.

Primers	Primers’ Sequences (5′-3′)	Annealing Temperature	Product Length (bp)
*p38-P4-F*	GCTACTTGAGCTGACTTACCTTG	57 °C	387 bp
*p38-P4-R*	ACAAACAAATTCTCAGTCTTTACAT		
*p38-P5-F*	ACAGTGAGCAAATCAGCAGTC	59 °C	352 bp
*p38-P5-R*	TCTCCCCCTGATGGTCTGAG		
*p38-P6-F*	CCAGAATTAGATCCAAGCTTTC	59 °C	217 bp
*p38-P6-R*	ACCTATGCAGGTCTACTATGCC		
*p38-P7-F*	AATACCTTAAGGCATAGTAGACC	58 °C	278 bp
*p38-P7-R*	CCAAGTATAAAAGAGGAGAAAAGATA		
*p38-P8-F*	TATCTTTTCTCCTCTTTTATACTTGG	56 °C	268 bp
*p38-P8-R*	ACCAACAACCACTTTATTTTATTTC		
*p38-P9-F*	TTCAGGTCTCCACGTTATTTGTTC	57 °C	382 bp
*p38-P9-R*	AATTCGTAAATAAATGTAGCTG		
*p38-P10-F*	AATACCTTAAGGCATAGTAGACCTGC	59 °C	669 bp
*p38-P10-R*	TGAATTTTCCGTGTGAATGGGAC		
*p38-P11-F*	CAGGACGTGACCTCTGATGC	57 °C	612 bp
*p38-P11-R*	TTCAGGTCTCCACGTTATTTGTTC		

## Data Availability

Data is contained within the article or Appendix A.

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
