# Peer review of "Non-Additive and Asymmetric Allelic Expression of p38 mapk in Hybrid Tilapia (Oreochromis niloticus ♀ × O. aureus ♂)"

_animals, 2024, doi:10.3390/ani14020266_

Round 1
Reviewer 1 Report (New Reviewer)
Comments and Suggestions for Authors
The present study reported the sequence of p38 mapk in Nile tilapia, blue tilapia and their hybrid individuals to find heterosis through interspecies crossing. The authors used SNP loci for the study and found eight specific SNP sites and two putative variations.
Overall the manuscript is good and seems to be revised once (the text is occasionally yellow-highlighted). The authors should consider the following issues:
- Keywords: use “hybridization” as a keyword.
- The text has lost its integrity, when the authors added some texts as revision (highlighted texts). The authors are invited to read the text again and polish it. For example line 316-336: the inserted text (highlighted) shows no connection to the subsequent sentence.
- I am wondering what was replication in this study? Fish pond or fish individuals? As the authors could not treat the fish one by one (feeding, environmental conditions …), the individuals cannot be considered as REPLICATION. Please clarify.
- The authors must state the statistics in more clear way. At present, we know only they used one way ANOVA and Duncan, but it is not clear what was analyzed by these tests.
- The authors must clarify if the b-actin of each individual was used as the reference gene, or average b-actin expression of one group of fish (Nile tilapia, blue tilapia or hybrid).
- The conclusion should be re-structured. It is more the methods and results, rather than concluding the results and potential application of the results.
Author Response
The present study reported the sequence of p38 mapk in Nile tilapia, blue tilapia and their hybrid individuals to find heterosis through interspecies crossing. The authors used SNP loci for the study and found eight specific SNP sites and two putative variations.
Overall the manuscript is good and seems to be revised once (the text is occasionally yellow-highlighted). The authors should consider the following issues:
On behalf of all the contributing authors, I would like to express our sincere appreciations of your letter and reviewers' constructive comments concerning our article entitled " Nonadditive and Asymmetric Allelic Expression of mapk14a 2 Gene in Hybrid Tilapia, Oreochromis niloticus ♀ × O. aureus ♂"(Manuscript No: 2717353). These comments are all valuable and helpful for improving our article. According to the associate editor and reviewers' comments, we have made extensive modifications to our manuscript and supplemented extra data to make our results convincing. In this revised version, our response is changes to the manuscript are given in the red text our responses are shown in blue font.
Keywords: use “hybridization” as a keyword.
The author’s answer: As suggested by the reviewer, we have changed the keyword to “hybridization”.
The text has lost its integrity, when the authors added some texts as revision (highlighted texts). The authors are invited to read the text again and polish it. For example line 316-336: the inserted text (highlighted) shows no connection to the subsequent sentence.
The author’s answer: We tried our best to improve the manuscript and made some changes to the manuscript. These changes will not influence the content and framework of the paper. And here we did not list the changes but marked in red. For example, line 316-336, we have added “Each of these examples identifies specific SNPs associated with growth and resistance by means of SNP molecular markers, which can be useful in fish breeding.” to make it relevant to the sentence that follows.
I am wondering what was replication in this study? Fish pond or fish individuals? As the authors could not treat the fish one by one (feeding, environmental conditions …), the individuals cannot be considered as REPLICATION. Please clarify.
The author’s answer: We think this is an excellent suggestion. It was an oversight on my part not to clearly describe the duplicates, in this project we duplicated the individual fish.
The authors must state the statistics in more clear way. At present, we know only they used one way ANOVA and Duncan, but it is not clear what was analyzed by these tests.
The author’s answer: Thank you for your question. Here is the problem with our expression, we have used one way ANOVA and Duncan here in expressing the analysis. And here we did not list the changes but marked in red in the revised paper.
The authors must clarify if the b-actin of each individual was used as the reference gene, or average b-actin expression of one group of fish (Nile tilapia, blue tilapia or hybrid).
The author’s answer: We sincerely thank the reviewer for careful reading. In this paper, b-actin was used as the reference gene for each individual.
The conclusion should be re-structured. It is more the methods and results, rather than concluding the results and potential application of the results.
The author’s answer: Thanks for your suggestion. We have done our best to structure the conclusions differently. And here we did not list the changes but marked in red in the revised paper. We appreciate for reviewer warm work earnestly and hope that the correction will meet with approval.
Reviewer 2 Report (New Reviewer)
Comments and Suggestions for Authors
See attached file.

Some phrasing are a bit strange and should be revised.
See comments for details.
Author Response
In the manuscript entitled “Nonadditive and assymetric allelic expression of p38 mapk in hybrid Tilapia, Oreochromis nicoticus ♀ × O. aureus ♂” the authors investigated the genomic variation in the gene of p38 mapk along with allele specific expression in hybrids offspring between two Tilapia species. The authors found no non-synonymous SNPs within the gene but they observed nonadditive expression in hybrids along with allele specific expression. Specifically, the allele of Nile Tilapia was more expressed than the allele originating from the Blue Tilapia.
I found the study interesting since there is not that many studies focusing on hybrids transcriptomic machinery and this could shed some light on the role of cis and trans-acting elements in this context. Specifically, the part on allele specific expression was really interesting and convincing.
However, I found that sometime the manuscript suffered from too general/vague statements from the literature to argue in favor of the authors ‘claims regarding their results. Hence some strong claims are not supported by similarly strong evidences to my opinion. I think notably about the claim regarding the role of the identified SNPs as impacting the regulation or the expression of the gene. I found it difficult to link the number of SNP found to what is observed in term of allele expression, since all of them are synonymous and there are no further evidences in that direction beside references on other model quite far from fish (pig and humans).
Similarly, the fact that hybrids are crossed in only one direction prevent from distinguishing species- specific effect from maternal/paternal effects. While it is not an issue per se I would still expect more nuanced conclusions and this bias to be discussed.
I would also appreciate to see in supplementary material some direct estimation of allele expression along with statistical significance measures in addition to what is displayed in Figure 4.
Finally, considering the claim about the discovery of marker to distinguish the two species, I would like to see further data on a more extended spatial scale to ensure that the markers are stable and conserved within at least part of each species range.
Response: We appreciate your letter and the constructive comments from the reviewers regarding our article (Manuscript No: 2717353). These comments are valuable and helpful for improving our article. Based on the associate editor and reviewers' feedback, we have extensively modified the manuscript and included additional data to strengthen our results. Addressing the issue you raised, we have incorporated relevant experiments to validate the activity of the SNP promoter region at a later stage, as detailed in Section 3.4. Additionally, we have revised the entire paper in line with your suggestions, ensuring the conclusions are more precise, and have included additional relevant literature. In this revised version, we have marked the changes in red.
Minor comments per section:
Introduction
L 44-45: Heterosis refers to the higher fitness of hybrids compared to their parents, while this can be expressed as faster growth rate this is not necessary always the case. Please rephrase in “Heterosis can be characterized by”
The author’s answer: We think this is an excellent suggestion. Line 43-44, we've rephrased the sentence as " Heterosis can be characterized by the ability to produce faster growing offspring."
- 48: Same as above, also please clarify this statement.
The author’s answer: We think this is an excellent suggestion. Line 47-48, we've rephrased the sentence as " Heterosis can be characterized by the ability to produce more adaptable offspring."
L.57-58: please provide a reference or nuance this statement. Change in gene expression is one of the current hypotheses to explain heterosis as observed in first generation hybrid but we are still at the stage of evidence accumulation.
The author’s answer: We sincerely appreciate the valuable comments. Line 57, we have checked the literature carefully and added relevant literature. (Reference 8, 9)
Reference 8: Gene expression analyses in maize inbreds and hybrids with varying levels of heterosis.
Reference 9: Transcriptome analysis reveals the molecular mechanisms of heterosis on thermal resistance in hybrid abalone.
L.60: Some studies tend to show association between non-additive expression and heterosis but this is not always the case (see for instance Guo et al., 2006). Similarly, non-additive expression of genes is not specific of heterosis in the sense that this expression pattern has been associated with hybrid incompatibilities as well.
The author’s answer: Please accept our sincere for our oversight. We value your advice and we are grateful that you brought this issue to our attention. In the text we have amended the statement here to " Non-additive gene expression is one of the factors contributing to heterosis".
L.82 Please defined abbreviated terms (such as GH here) at first occurrence.
The author’s answer: We sincerely apologize for our oversight and appreciate your feedback. We have already defined abbreviated terms at first occurrence in the text, such as GH means growth hormone.
L.96-99 Please reorder the sentence: In fish, such as rock bream, Atlantic salmon etc, previous studies have focused…
The author’s answer: We sincerely thank the reviewer for careful reading. As suggested by the reviewer, line88-101, we have rearranged the order of the sentences.
- 101 Please review the phrasing. An analysis cannot find something (it can reveal and the authors can find for instance).
The author’s answer: We sincerely apologize for our oversight and appreciate your feedback. Line 103, we have reworded ourselves in the article.
L.106 I find it a bit clumsy to finish the introduction on an example on rats. I would suggest to remove this part and instead add more details about what has been done in this study, the hypothesis tested and the expectations according to the current litterature on Tilapia.
The author’s answer: We think this is an excellent suggestion. We've removed the rat example from the text and added more details about what has been done in this study, the hypothesis tested and the expectations according to the current literature on Tilapia.
Material and Methods
- 123 so these hybrids resulted from one single cross direction? Why the mirrors hybrids have not been produced as well? Is that because they are not as viable?
The author’s answer: Thank you for your question. As the reviewer's idea, in practical production we found that another cross direction to obtain individual was more difficult and less successful, and we only made one single cross direction.
- 152 why reducing feeding qt 45d?
The author’s answer: Thank you for your question. We were really sorry for our careless mistakes. The trial was a 90d trial, and we sampled and measured some indices at 45d and 90d, but these data are not shown in the manuscript.
- 170 Where the 20 individuals from the same population/location? Is there any evidence that the SNPs identified would be conserved across different population of the two species?
The author’s answer: Thank you for your question. Although in this study we show a sample of 20, we actually randomized a larger sample later and the results are consistent with this, which we have modified for accuracy of presentation. Additionally, as China's largest tilapia breeding research base, our base currently have the most comprehensive collection of tilapia populations in the whole country. Therefore, the accuracy of our research findings is sufficient.
- 187 I don’t understand how a SNP can serve as reference. Can you please provide more information here and rephrase if necessary. Also, how long are the amplified fragments?
The author’s answer: Thank you for your question. We were really sorry for our careless mistakes. Line 185, SNP is a site-specific locus, which we have modified in the revised manuscript. Amplified fragment lengths have been expressed in Table 2.
- 210 Please rephrase. Were calculated following the equation provided in Figure 1?
The author’s answer: Thanks for your suggestion. Line 206, this has been restated in the original text and calculated according to the formula provided in Figure 1.
- 228 Please clarify. A SNP is a one-point mutation in a sequence. Do you mean that you investigated if the SNPs altered the function of the gene?
The author’s answer: Thank you for your question. Here in the text it is not a change in function, but SNP changes in the promoter region that may affect gene expression.
- 232-235: please provide more information about the group compared and the model used.
The author’s answer: We sincerely thank the reviewer for careful reading. We used the analytical approach in section 3.3.
Results
- 244 I might have missed it, but what is the reference used for calling the SNPs (Nile Tilapia or Blue Tilapia)? Also, what does it mean when a group does not have a SNP, is that that they have the same nucleotide as the reference?
The author’s answer: Thank you for your question. The reference genome for this article is the Nile tilapia. In addition, the absence of a SNP within a population(group) indicates a uniform genotype at that specific locus among its members.
- 249 Because these SNPs are all synonymous mutation. I am even more worried about the idea of using them as diagnostic marker to identify species. Indeed, they could more easily than non- synonymous mutations evolve by drift and be lost and/or differ from one population to the other.
The author’s answer: Thanks for your suggestion. Our population analysis of the parents determined that SNP differences did exist between the two species, and thus the validity of the typing analysis of the progeny obtained by crossing is considered acceptable, but the reminder from the reviewer is very important, and our next step is to consider future related studies to carry out analyses from more than one locus, but in this study from the perspective of funding we only did one locus.
- 261 The title is the same as the previous paragraph.
The author’s answer: We sincerely thank the reviewer for careful reading. We are sorry for our carelessness. Line 261, we have changed the title here to "Gene p38 allele-specific expression ".
- 281 It is not clear to me what data show the compensatory effect.
The author’s answer: Thanks for your suggestion. This is calculated from the data, and here is a part of the definition, meanwhile, we have a part of the experiments to supplement it in line 232-262.
- 303 I agree with this statement and it is more likely that those SNPs in the promoter region are linked to the change in allele expression in hybrid rather than the synonymous SNPs previously identified.
The author’s answer: We think this is an excellent suggestion. Based on your comments, line 333-335, we have made a correction, which reads " The allele-specific expression observed in HY from different parental sources may be due to SNPs in the promoter region associated with changes in allele expression in hybridization."
Discussion
- 312-313 It is a rather vague statement, all SNPs are not behaving the way the authors state it. As a punctual mutation whether it is heritable and/or stable greatly depend on the location of the SNP in the genome. Also, a lot of mutations are just neutral or silent.
The author’s answer: Thank you for your question. It is true that whether a SNP works or not depends very much on the location of the SNP on the genome, and in fact, during the experiment we screened for relatively stable SNP. In the meantime, line 353-356, we have rewritten this section to read " For example, polymorphisms in promoter regions affect promoter activity and gene expression, and SNP sites affect the methylation level of promoter regions, causing changes in the translation process, which can lead to changes in individual phenotypes".
- 339 hybrid repeated twice.
The author’s answer: We sincerely thank the reviewer for careful reading. Based on your comments, line 384-385, We have changed the presentation.
- 341 Some research have shown.
The author’s answer: We sincerely thank the reviewer for careful reading. We have changed “Research has demonstrated” to “Some researches have shown”.
- 355-356 Again, I am not opposed to this hypothesis, thought I would expect more nuances if no further evidence for such a mechanism going on in the tilapia hybrids. Also, the other type of SNP in the promoter region had a more direct potential for impacting downstream gene expression.
The author’s answer: Thank you for your question. In fact, we followed up with experiments to show that changes in SNPs in promoter regions affect gene expression, as described in section 3.5. And here we did not list the changes but marked in red in the revised paper.
- 360 what do the authors exactly refer to when talking about genetic elements?
The author’s answer: We sincerely thank the reviewer for careful reading. We are sorry for our carelessness. Here is an error in our formulation, we wanted to express that transcription can be affected by transcription factors. We did not list the changes but marked in red in the revised paper.
L 403-416 This paragraph should be revised to my opinion. The study on coffee is just an example and does not really explain directly what occur in Tilapia. I think the different points could be brought in a better way separating better results found in the literature and specific results in tilapias as provided in this study.
The author’s answer: We think this is an excellent suggestion. Line 455-458, We have rewritten this section based on the reviewers' suggestions as follows" The mechanism can also be applied in studying the mechanism of non-additive expression in hybrid tilapia, as the compensatory effects of cis + trans acting elements from males and females can determine the genetic pattern of p38 ELOD expression in hybrid tilapia."
References
Please ensure that species names are italicized in the references.
The author’s answer: We sincerely thank the reviewer for careful reading. Based on your comments, we have changed all species names in the references to italics and highlighted them in yellow.

Round 2
Reviewer 1 Report (New Reviewer)
Comments and Suggestions for Authors
Accepted
Author Response
We appreciate your letter and the constructive comments from the reviewers regarding our article (Manuscript No: 2717353).Thank you very much for your endorsement of our article. We wish you all the best in your work and a happy life!
Reviewer 2 Report (New Reviewer)
Comments and Suggestions for Authors
I have reviewed the revised version of the MS “Nonadditive and assymetric allelic expression of p38 mapk in hybrid Tilapia, Oreochromis nicoticus ♀ × O. aureus ♂” according the comments previously made.
I found the revised version of the MS more convincing and most of my previous comments have been considered, although some answers to my comments were only provided in the response to reviewer and did not lead to precisions added in the MS (eg. For my previous question: Is there any evidence that the SNPs identified would be conserved across different population of the two species?)
I have not seen an answer or precision added regarded my previous comment on the potential confusion between species specific effect and maternal/paternal effects.
Similarly, I don’t know if I am missing the supplementary during the review process, but otherwise I would like to re-iterate my comment about providing direct estimation of allele expression along with statistical significance in supplementary. This is essential to judge the reliability of the results.
I also would like to see brief description of the methods used, such as for the dual luciferase activity assay, but also for other occurrences when the authors mentioned following the method of another study (eg. L. 184 and 207).
Finally, I would recommend careful proof reading for correction of the phrasing since some sentences are either quite long or not always grammatically correct.
Minor comments
L. 98 have focused twice
L. 107 please correct the phrasing. I suggest: this study rather than experiment and assessment rather than study.
L. 107-110: please consider making several shorter sentences to increase clarity
L. 2051 nega-tive splitted, this typo occurred in various places with other words (eg. 243, 2470 etc)
L. 232 Please provide a brief description of the methodology
Fig 6 panel B is too small and difficult to read
L. 351 I am still not convinced by how the discussion is beginning with the direct focus on how SNP can alter gene expression. While this is likely the case, SNP were primarily studied for their potential of changing a sequence, of a neutral mutation to study demographic changes. It is only later on that they were also considered for the potential impacts on gene regulation through for instance methylation.
I don’t think the added sentence should be integrated that early in the discussion and would recommend to rather start with more general statements.
L. 455 Mechanism used twice in the sentence.
Comments on the Quality of English LanguagePhrasing is sometimes a bit akward and could be improved (eg. long not very clear sentences).
some typos are also present.
Author Response
I have reviewed the revised version of the MS “Nonadditive and assymetric allelic expression of p38 mapk in hybrid Tilapia, Oreochromis nicoticus ♀ x O. aureus♂” according the comments previously made.
I found the revised version of the MS more convincing and most of my previous comments have been considered, although some answers to my comments were only provided in the response to reviewer and did not lead to precisions added in the MS (eg. For my previous question: Is there any evidence that the SNPs identified would be conserved across different population of the two species?)
I have not seen an answer or precision added regarded my previous comment on the potential confusion between species specific effect and matemal/patemal effects.
Similarly, I don't know if I am missing the supplementary during the review process, but otherwise I would like to re-iterate my comment about providing direct estimation of allele expression along with statistical significance in supplementary. This is essential to judge the reliability of the results.
I also would like to see brief description of the methods used, such as for the dual luciferase activity assay, but also for other occurrences when the authors mentioned following the method of another study (eg. L.184 and 207).
Finally, I would recommend careful proof reading for correction of the phrasing since some sentences are either quite long or not always grammatically correct.
We appreciate your letter and the constructive comments from the reviewers regarding our article (Manuscript No: 2717353). All of us authors have carefully read the comments that you have given us, and have discussed and revised each of these issues. We have briefly explained the methodology used in the revised version, and we have done our best to revise the wording and grammar of the article. In addition, we have resubmitted a new manuscript in the revised state, with the revisions highlighted in red. If there are any incorrect answers or questions in the manuscript, please do not hesitate or let us know.
Respond to the editor’ comments:
1.Is there any evidence that the SNPs identified would be conserved across different population of the two species?
The author’s answer: Thank you for your question. In fact, in our team's previous work, we performed genetic analyses on hybrid tilapia parents and our data showed that all of these SNPs are conserved in our experimental fish including Blue tilapia and Nile tilapia.
- Supplementary material on allele expression and statistical significance.
The author’s answer: We sincerely thank the reviewer for careful reading. We placed material related to allele expression and statistical significance in Table S1.
Minor comments
L.98 have focused twice
The author’s answer: Thank you for your question. We were really sorry for our careless mistakes. Line 98-99, we have removed the duplicate of "have focused" here.
L.107 please correct the phrasing. I suggest this study rather than experiment and assessment rather than study.
The author’s answer: We sincerely thank the reviewer for careful reading. Based on the reviewers' suggestions, line 107, We have replaced “in this experiment” with “in the study” and “performed” with “assessed”.
L.107-110: please consider making several shorter sentences to increase clarity
The author’s answer: We sincerely appreciate the valuable comments. Following the reviewer's suggestion, line 107-110, we have shortened the sentence as follows “Therefore, in the study, we analysed the sequence and assessed the allele-specific expression pattern of p38 in hybrid tilapia, and explored the regulation of p38 expression by cis- and trans-acting elements in hybrid tilapia with a view to demonstrating that p38 can promote fish growth.”
L.205 1 negative splitted, this typo occurred in various places with other words (eg.243,2470 etc)
The author’s answer: We sincerely thank the reviewer for careful reading. We were really sorry for our careless mistakes. "nega-tive splitted" is a misnomer that we have corrected throughout the text. (Line 244 and 246)
L.232 Please provide a brief description of the methodology
The author’s answer: We sincerely appreciate the valuable comments. Based on the reviewers' comments, line 232-250, we did our best to shorten the description of the methodology.
Fig 6 panel B is too small and difficult to read
The author’s answer: We sincerely appreciate the valuable comments. In the revised manuscript, we have reworked Panel B of Figure 6 and added relevant notes to make it easier to read.
L.351 I am still not convinced by how the discussion is beginning with the direct focus on how SNP can alter gene expression. While this is likely the case, SNP were primarily studied for their potential of changing a sequence, of a neutral mutation to study demographic changes. It is only later on that they were also considered for the potential impacts on gene regulation through for instance methylation.I don't think the added sentence should be integrated that early in the discussion and would recommend to rather start with more general statements.
The author’s answer: We sincerely appreciate the valuable comments. Line 339-344, we have rephrased this as follows “SNP is a common genetic molecular marker developed in recent years, which has the characteristics of strong heritability and high stability. Initially, the study of SNPs was aimed at understanding their potential for sequence alteration. However, as molecular markers and their detection techniques advance, the perspective shifted to considering the potential impact of SNPs on gene regulation, such as through methylation processes. Therefore, SNPs are also widely used in various genetic research fields.”
L.455 Mechanism used twice in the sentence.
The author’s answer: We sincerely thank the reviewer for careful reading. We were really sorry for our careless mistakes. In the revised draft, line 440-443, we have deleted the duplicate " mechanism ".
This manuscript is a resubmission of an earlier submission. The following is a list of the peer review reports and author responses from that submission.
Round 1
Reviewer 1 Report
Comments and Suggestions for Authors
Inter-specific hybridization is very common in plants and animals, however, the underlying mechanism of heterosis in hybrids is not resolved. The hybrid tilapia, Oreochromis niloticus × Oreochromis aureus, is commonly appplied in tilapia production, with advantages such as high male ratios, rapid growth, and good disease resistance. This manuscript provides some valuable insights into the gene expression variation in inter-specific hybridization of tilapia. The study demonstrated non-additive and specific expression of the mapk14a gene in hybrid tilapia, shedding light on the gene regulation mechanism in the inter-specific hybridization process.
Although the experimental layout is acceptable. it would be beneficial to provide a clearer description in the methods, it also be beneficial with English language editing to improve readability.
Line 111: Add the approval code and date of the "Animal Care and Use Committee of Nanjing Agricultural University".
Line 121: Provide the details of the commercial diet, such as proximate components.
Lines 133-135: Were different tilapia groups kept in same unit or separately? Provide a clear description.
Lines 191-193: In Table 1, primer information for β-actin and mapk14a is missing.
Line 228: Please add the number of replications for the different groups involved in the experiments.
Line 247: In Figure 2A, the UTR is part of the exon, but the guide line added by the authors seems to separate the UTR from the exon. In fact, the UTR should be separated from the CDS (as marked by different colors in Figure 2A). Exon 1 and exon 12 in this study are a combination of the aforementioned parts, please revise accordingly. Additionally, the guide line for exon 6 is missing.
Line 247: In Figure 2B, improve the display to make it more readable.
Line 276: What do the added plus signs and their number represent for the different tilapia in Figure 5? Please explain within the caption and statistical analysis section. Also, include a description of the meaning of the different shapes and marking lines within the caption.
Line 283: The phrase "loci between the groups" does not clearly describe the polymorphisms of the loci. It may lead the reader to believe that there are polymorphisms in both groups, but the results show that "NL has more possible TFBS than AR," suggesting that the polymorphisms between NL and AR are inconsistent. Please provide details, such as genotypes and frequencies, for the same locus in different tilapia (this information can be presented in an additional table). There is a similar presentation issue in 3.1, the authors have added some notes to make the distribution of SNPs between groups clearer. I suggest adding information about the polymorphisms at each locus.
Reviewer 2 Report
Comments and Suggestions for Authors